

# Lyophilized cell-free supernatants of *Lactobacillus* isolates exhibited antibiofilm, antioxidant, and reduces nitric oxide activity in lipopolysaccharide-stimulated RAW 264.7 cells

Phoomjai Sornsenee[1], Moragot Chatatikun[2,3,4], Watcharapong Mitsuwan[5,6,7], Kantapich Kongpol[2,4], Nateelak Kooltheat[2,4], Sasirat Sohbenalee[2], Supawita Pruksaphanrat[2], Amron Mudpan[2] and Chonticha Romyasamit[2,3,5]

[1] Department of Family and Preventive Medicine, Faculty of Medicine, Prince of Songkla University, Hat Yai, Songkhla, Thailand

[2] Department of Medical Technology, School of Allied Health Sciences, Walailak University, Thasala, Nakhon Si Thammarat, Thailand

[3] Center of Excellence Research for Meliodosis (CERM), Walailak University, Thasala, Nakhon Si Thammarat, Thailand

[4] Research Excellence Center for Innovation and Health Product, Walailak University, Thasala, Nakhon Si Thammarat, Thailand

[5] Research Center of Excellence in Innovation of Essential Oil, Walailak University, Thasala, Nakhon Si Thammarat, Thailand

[6] One Health Research Center, Walailak University, Thasala, Nakhon Si Thammarat, Thailand

[7] Akkhraratchakumari Veterinary College, Walailak University, Thasala, Nakhon Si Thammarat, Thailand

Corresponding author
Chonticha Romyasamit,
chonticha.ro@wu.ac.th

## ABSTRACT

**Background**. Probiotics can release bioactive substances known as postbiotics, which can inhibit pathogenic microorganisms, improve immunomodulation, reduce antioxidant production, and modulate the gut microbiota.

**Methods**. In this study, we evaluated the *in vitro* antimicrobial effects, antioxidant activity, and anti-inflammatory potential of 10 lyophilized cell-free supernatants (LCFS) of *Lactobacillus* isolates. LCFS was obtained via centrifugation and subsequent lyophilization of the supernatant collected from the culture medium ofeach isolate. The antibacterial and antibiofilm activities of the LCFS were determined using broth microdilution. The antioxidant potential was evaluated by measuring the total phenolic and flavonoid contents and 2,2-Diphennyl-1-picrylhydrazyl (DPPH) and 2,2'-azinobis(3-ethylbenzothiazoline-6-sulfonic acid) radical cation (ABTS$^+$) radical scavenging activities.

**Results**. All the isolates were able to inhibit the four tested pathogens. The isolates exhibited strong antibiofilm activity and eradicated the biofilms formed by *Acinetobacter buamannii* and *Escherichia coli*. All the prepared *Lactobacillus* LCFS contained phenols and flavonoids and exhibited antioxidant activities in the DPPH and ABTS$^+$ radical scavenging assays. The MTT (3-[4,5-dimethylthiazol-2-yl]-2,5 diphenyl tetrazolium bromide) assay revealed that LCFS was not cytotoxic to RAW 264.7 cells. In addition, the ten *Lactobacillus* LCFS decreased the production of nitric oxide.

**Conclusions**. All the isolates have beneficial properties. This research sheds light on the role of postbiotics in functional fermented foods and pharmaceutical products.

Further research to elucidate the precise molecular mechanisms of action of probiotics is warranted.

# INTRODUCTION

The term "probiotics" refers to living or dead microorganisms that confer health benefits to a host when administered in adequate amounts (*Hotel & Cordoba, 2001*). Probiotic microorganisms exert their benefits through two mechanisms: direct effects on living cells and indirect effects involving the production of several metabolites (*Vinderola et al., 2007*). The most frequently used probiotic microorganisms are lactic acid bacteria (LAB) such as *Lactobacillus* spp., *Lactococcus* spp., *Carnobacterium* spp., *Enterococcus* spp., *Streptococcus* spp., *Pediococcus* spp., and *Propionibacterium* spp. (*Pinto et al., 2020*; *Sornsenee et al., 2021*). Generally, *Lactobacillus* spp. are the most popular probiotic microbes owing to their "Generally Recognized As Safe (GRAS)" status and their regulation by the US Food and Drug Administration (FDA) for human and animal consumption (*FAO/WHO, 2002*; *Sornsenee et al., 2021*). For example, *L. acidophilus* CL1285, *L. casei* LBC80R, *L. rhamnosus* CLR2 (Bio-K Plus International Inc, Laval, Quebec, Canada), *L. acidophilus* (La-5®), and *Bifidobacterium lactis* (BB-12®) (Pharma Nord, Nederland), have been used as probiotics in pharmaceutical and diet supplements (*Organization, 2002*).

The beneficial effects of *Lactobacillus* as probiotics are not limited to the health of the gastrointestinal tract (GIT) and extend to conditions such as diabetes, obesity, hyperlipidemia, cancer, dementia, Crohn's disease, and constipation (*Plaza-Diaz et al., 2019*). Probiotics produce organic acids (acetic acid, propionic acid, and lactic acid), aromatic compounds, diacetyl, hydrogen peroxide, antimicrobial substances, bacteriocins, and other unknown metabolites (*Barzegari et al., 2020*; *Bermudez-Brito et al., 2012*; *Cremon et al., 2018*) that can inhibit several pathogens such as *Clostridium difficile* (*Shahrokhi & Nagalli, 2020*), *Vibrio parahaemolyticus* (*Behera, Ray & Zdolec, 2018*), carbapenem-resistant *Escherichia coli* (*Chen et al., 2019*), *Klebsiella pneumoniae* (*Chen et al., 2019*), *Listeria monocytogenes* (*Kariyawasam et al., 2020*), *Staphylococcus aureus* (*Melo et al., 2016*), *Salmonella enteritidis* (*Sornsenee et al., 2021*), and *Helicobacter pylori* (*Ji & Yang, 2021*). Probiotics can lower cholesterol levels, boost the immune system, promote the secretion of immunoglobulin IgA, serve as antioxidants, exhibit antidiabetic properties, and suppress inflammation (*AlKalbani, Turner & Ayyash, 2019*; *De Marco et al., 2018*; *Singhal et al., 2019*; *Xu et al., 2021*). Several studies have shown that *Lactobacillus* can inhibit biofilm formation by many pathogens (*Carvalho et al., 2021*; *Gómez et al., 2016*; *Ji & Yang, 2021*; *Shaaban et al., 2020*). Other reports have shown that metabolites produced by probiotics have antivirulence activity (*Stefania et al., 2017*).

Members of the genus *Lactobacillus* are gram-positive bacteria, aerotolerant anaerobes or microaerophilic, rod-shaped, and non-spore-forming, with low DNA G+C content

(*Klaenhammer et al., 2005*). This genus comprises 261 species as of March 2020, with extreme diversity at phenotypic, ecological, and genotypic levels (*Zheng et al., 2020*). We previously identified 10 *Lactobacillus* isolates from fermented palm sap collected from a local market in the Songkhla Province of Southern Thailand. All *Lactobacillus* isolates met the established criteria to qualify as potential probiotics, including resistance to gastrointestinal conditions, adherence to human intestinal cells, and susceptibility to transmissible antibiotics. These isolates possessed antimicrobial activity against a wide range of pathogens (*Sornsenee et al., 2021*). From these data, 10 *Lactobacillus* isolates are promising potential candidates for use as probiotic applications as functional foods and pharmaceutical products. However, we still lack information about the antibiofilm, antioxidant, and anti-inflammatory activities of *Lactobacillus* isolates. Thus, the present work aimed (i) to evaluate the antibacterial and antibiofilm activities of lyophilized cell-free supernatants (LCFS) of *Lactobacillus* against pathogens, (ii) to evaluate the total phenolic and flavonoid contents and free-radical-scavenging activities, and (iii) to evaluate the toxicity of the cell-free supernatants (CFS) and their anti-inflammatory activity using RAW 264.7 cells.

## MATERIALS & METHODS

### Microorganisms and culture conditions

Ten *Lactobacillus* isolates, including *L. paracasei* (T0601, T0602, T0603, T0901, T0902, T1301, T1304, and T1901), *L. fermentum* (T0701), and *L. brevis* (T0802), were isolated from fermented palm sap collected from a local market in the Songkhla Province of Southern Thailand and characterized as potential probiotics in our previous study. These isolates were used in the present study (*Sornsenee et al., 2021*). First, they were grown in de Man, Rogosa and Sharpe (MRS) broth (HiMedia, Mumbai, India) at 37 °C for overnight. After that, all isolates were stored at −80 °C in 30% (v/v) glycerol (Sigma, Steinheim, Germany).

Three reference strains, *E. coli* DMST4212, *A. baumannii* DMST 2271, and *S. aureus* DMST 2928, obtained from the Department of Medical Sciences Thailand (DMST), were used in this study. One clinical isolate, methicillin-resistant *S. aureus* (MRSA), was identified using matrix-assisted laser desorption ionization time-of-flight mass spectrometry/MS mass spectrometry. These strains were cultured on trypticase soy (TSA) agar (HiMedia, Mumbai, India), and the agar plates were incubated at 37 °C for 18 h under aerobic conditions. The colonies were transferred to trypticase soy broth (HiMedia, Mumbai, India) and incubated at 37 °C for 18 h. Each strain was stored at −80 °C in brain heart infusion broth with 30% glycerol until further use.

### Preparation of CFS

CFS were prepared according to *Melo et al. (2016)* with slight modifications. Briefly, each *Lactobacillus* isolate was cultured in 100 mL of MRS broth and incubated at 37 °C for 18 h under anaerobic conditions. The supernatant was obtained by centrifugation ($\times 6,000$ g, 10 min, 4 °C). The centrifuged supernatant was passed through a sterile 0.22 $\mu$m-pore-size filter unit (Sigma, Steinheim, Germany). The filtrate was collected for freeze-drying.

## Lyophilization

CFS of each *Lactobacillus* isolate and MRS medium without *Lactobacillus* (MRS control) were frozen at −80 °C for 24 h. The samples were lyophilized (Lyophilization Systems, Inc, USA) from −40 ° C to −30 °C, 0.2 mbar. The entire freeze-drying process was performed in 24 h, and the freeze-dried powders were stored at −20 °C. They were then rehydrated with sterile deionized water prior to use.

## Determination of minimum inhibitory concentration (MIC) and minimal bactericidal concentration (MBC)

The antibacterial activities of each LCFS against the four pathogenic bacteria were assessed using the method of microdilution in 96-well plates according to the Clinical and Laboratory Standards Institute (CLSI) 2021 guidelines (*CLSI, 2021*). Serial dilution was performed starting with 100 mg/mL of lyophilized CFS of *Lactobacillus* in Mueller Hinton broth (MHB) (HiMedia, Mumbai, India). The bacterial suspension ($5 \times 10^5$ CFU/mL) was inoculated into each well, and the plates were incubated at 37 °C for 18 h. Then, resazurin (Sigma, Steinheim, Germany) was used to determine the MIC values. The MIC was defined as the lowest concentration that completely inhibited the bacterial growth, which presented as a blue color (*Hussain et al., 2011*). The MBC was determined using the extract that yielded significant MIC values by dropping the culture onto TSA plates. The entire experiment was performed three times with three independent repetitions.

## Biofilm inhibition assay

The effects of LCFS of *Lactobacillus* on biofilm formation of *E. coli* DMST4212 and *A. baumannii* DMST 2271 were performed following a method that was modified from published by *Yang et al. (2021)*. Briefly, overnight cultures of *E. coli* DMST4212 and *A. baumannii* DMST 2271 were suspended in MHB to a cell density of $5 \times 10^5$ CFU/mL and then inoculated into 96-well plates supplemented with $1\times$ MIC and $2\times$ MIC of CFS of *Lactobacillus*. The plates were incubated at 37 °C for 24 h under aerobic conditions. Then, the medium was removed, the biofilms were washed with phosphate-buffered saline (PBS) (pH 7.4) three times, and fixed with 99% (v/v) methanol (200 μL) for 15 min. The biofilm was stained with 0.1% (w/v) crystal violet solution (200 μL) for 10 min. The wells were rinsed four times with distilled water to remove excess dye. The biofilms were dissolved in 95% (v/v) ethanol and absorbance was measured at an optical density (OD) of 570 nm. Each test was performed in triplicate. The percentage of biofilm inhibition was calculated using the following equation:

Biofilm inhibition (%) = [(OD 570 of control well − OD 570 of treated well)/OD 570 of control well] × 100.

## Biofilm eradication assays

The effects of LCFS of *Lactobacillus* on the eradication of biofilms produced by *E. coli* DMST4212 and *A. baumannii* DMST 2271 were tested according to reported procedures of *Perumal & Mahmud (2013)* with slight modifications. Briefly, an overnight culture of each *E. coli* DMST4212 and *A. baumannii* DMST 2271 was added to a 96-well microtiter plate and incubated at 37 °C for two days to allow the development of a biofilm. Then,

the wells were rinsed with PBS (pH 7.4) to remove non-adherent cells. The biofilms established for two days in each well were subsequently treated with $1\times$ MIC and $2\times$ MIC of CFS of *Lactobacillus* and incubated at 37 °C for 24 h. After incubation, the plates were removed, gently washed with PBS three times, and stained with 0.1% (w/v) crystal violet solution, as described previously, to determine the extent of biofilm inhibition. Each test was performed in triplicate. The percentage of biofilm eradication was calculated using the following equation:

Biofilm eradication (%) = [(OD 570 of control well − OD 570 of treated well)/OD 570 of control well] $\times$100.

## Determination of antioxidant activity
### Total phenolic content (TPC) assay
The Folin–Ciocalteu method was used to determine TPC, as described by *Chatatikun et al. (2020)* with some modifications. Briefly, LCFS of *Lactobacillus* was diluted in distilled water to a concentration of 50 mg/mL. Subsequently, 100 µL of 0.1 M $Na_2CO_3$ solution and 100 µL of 10% Folin–Ciocalteu reagent (Sigma-Aldrich, St. Louis, USA) were mixed in a well of a 96-well plate and incubated for 1 h. The absorbance was measured at 750 nm. A standard curve was plotted using gallic acid with a concentration range of 1.569–200 µg/mL. TPC was determined as gallic acid equivalents (GAE) in mg/g of lyophilized CFS of *Lactobacillus*.

### Total flavonoid content (TFC) assay
The TFC of the LCFS of *Lactobacillus* was determined using the aluminum chloride colorimetric method (*Chatatikun et al., 2020*). Briefly, 100 µL CFS of *Lactobacillus* or quercetin (1.56–100 µg/mL) was incubated with 100 µL of 2% $AlCl_3$ solution in methanol for 30 min at room temperature, and the absorbance was measured at 415 nm. The TFC was calculated from a calibration curve, and the result was expressed as mg quercetin equivalents (QE) per g of lyophilized CFS of *Lactobacillus*.

### 2,2-Diphennyl-1-picrylhydrazyl (DPPH) radical scavenging activity
The free-radical-scavenging activities of LCFS of *Lactobacillus* were measured using the DPPH assay with Trolox (Sigma-Aldrich, St. Louis, MI, USA) as the standard. This assay was performed according to the procedure previously described by *Chatatikun et al. (2020)* with some modifications. Briefly, 1,000 µg/ml of CFS of *Lactobacillus* (20 µL) or 1.56 to 100 µg/ml ascorbic acid standard in absolute ethanol was added to 180 µL of DPPH working solution. Then, the mixture was shaken and incubated in the dark for 30 min. The absorbance was read at 517 nm against a blank. The assays were done in triplicate. The DPPH scavenging activity was calculated using the following equation:

% Scavenging activity = 100 $\times$ (Abs of control − (Abs of sample − Abs of blank))/Abs of control. IC50, the concentration resulting in 50% inhibition of DPPH, was determined from a graph of free-radical-scavenging activity.

## ABTS$^+$ radical scavenging activity
ABTS*+ is generated by oxidation with a strong oxidizing agent (potassium persulfate). The reduction of a blue–green color of ABTS*+ free radical by donating hydrogen antioxidants

from LCFS is determined by the decrease of its absorbance. The ABTS$^+$ radical scavenging activity of LCFS of *Lactobacillus* was evaluated using an ABTS decolorization assay as published by *Chatatikun et al. (2020)* with modifications. Briefly, ABTS$^+$ was produced by mixing 7 mM ABTS and 2.45 mM potassium sulfate at a ratio of 2:3 (v/v). The ABTS$^+$ was stored in the dark at room temperature for 15 h until it was used. The ABTS$^+$ solution was diluted with methanol to reach an absorbance of $0.70 \pm 0.02$. Then, 20 µL of CFS of *Lactobacillus* were mixed with 180 µL of ABTS$^+$ solution and incubated for 45 min. The assays were done in triplicate. The percent inhibition of absorbance at 734 nm was calculated using the following equation:

% Scavenging activity = 100 × (Abs of control − (Abs of sample − Abs of blank))/Abs of control. IC50 was determined as the concentration resulting in 50% inhibition of ABTS$^+$

## Determination of anti-inflammatory activity
### Cell culture
RAW 264.7 cells, a mouse macrophage cells were kindly provided by Assoc. Prof. Dr. Potchanapond Graidist, Department of Biomedical Sciences and Biomedical Engineering, Faculty of Medicine, Prince of Songkla University, Hatyai, Songkhla, Thailand. RAW 264.7 cells were cultured in Dulbecco's Modified Eagle's Medium (DMEM; Gibco, Thermo Fisher Scientific, NY, USA) with 10% fetal bovine serum (Gibco) and 1% penicillin–streptomycin solution (Gibco, Thermo Fisher Scientific) at 37 °C in 5% $CO_2$. The RAW 264.7 cells were subcultured and plated at 80%–90% confluency.

### Cell viability assays
MTT assays were performed to assess the effect of LCFS of *Lactobacillus* on the viability of RAW 264.7 cells with modifications (*Khanna et al., 2020*). Briefly, RAW 264.7 cells were seeded onto 96-well microplates at $1 \times 10^5$ cells/mL and incubated at 37 °C in a 5% $CO_2$ incubator for cytotoxicity assays. The cells were then treated with CFS from *Lactobacillus* and incubated at 37 °C for 16 h. After incubation, supernatants were discarded and the cells were washed with PBS. A volume of 50 µL of 3-(4,5-dimethylthiazol-2-yl)-2,5-diphenyl tetrazolium bromide (MTT) solution (Sigma, MO, USA) (0.5 mg/mL in DMEM) was added to each well and incubated for 4 h in the dark after removing the treatment mixture from each well. The formazan crystals were dissolved by adding 100 µL of dimethylsulfoxide (DMSO) solution (Sigma, MO, USA). The OD was measured at 570 nm using a microplate reader. The experiment was repeated three times with triplicate samples. The percentage of cell viability was calculated using the following equation:

% cell viability = (OD of test/OD of untreated control) × 100

### Nitric oxide assays
To evaluate their anti-inflammatory activity, the LCFS of *Lactobacillus* were tested for their ability to reduce lipopolysaccharide (LPS)-induced nitric oxide (NO) generation in RAW 264.7 cells according to the method of *Khanna et al. (2020)* with slight modifications. Briefly, RAW 264.7 cells were seeded in a 24-well microplate and treated with 96.52 µg/L of LCFS of *Lactobacillus* with or without 1 µg/ml of LPS (Sigma-Aldrich, St. Louis, USA). RAW 264.7 cells treated with 1 µg/ml of LPS alone were used as the positive control. After

**Table 1** Minimal inhibitory concentration (MIC) and minimal bactericidal concentration (MBC) of LCFS of *Lactobacillus* on the four pathogens (*S. aureus,* MRSA, *E. coli, A. baumannii*).

| Isolates | Antimicrobial activity (mg/mL) | | | | | | | |
|---|---|---|---|---|---|---|---|---|
| | *S. aureus* | | MRSA | | *E. coli* | | *A. baumannii* | |
| | MIC | MBC | MIC | MBC | MIC | MBC | MIC | MBC |
| T0601 | 50 | >100 | 50 | >100 | 25 | >100 | 25 | >100 |
| T0602 | 25 | >100 | 25 | >100 | 25 | >100 | 25 | 100 |
| T0603 | 25 | >100 | 25 | >100 | 25 | >100 | 25 | >100 |
| T0701 | 25 | >100 | 50 | >100 | 50 | >100 | 25 | >100 |
| T0802 | 25 | >100 | 25 | >100 | 50 | >100 | 50 | >100 |
| T0901 | 25 | >100 | 25 | >100 | 25 | >100 | 25 | >100 |
| T0902 | ND | ND | ND | ND | 25 | >100 | 25 | >100 |
| T1301 | ND | ND | ND | ND | 25 | >100 | 25 | >100 |
| T1304 | ND | ND | ND | ND | 25 | >100 | 25 | >100 |
| T1901 | 25 | >100 | 25 | >100 | 50 | >100 | 25 | >100 |

**Notes.**
This test was performed in triplicate.
ND, Not detectable; MRSA, Methicillin-resistant *S. aureus*.

24 h of incubation at 37 °C in 5% $CO_2$, the nitric oxide production was measure by treating the supernatant with an equal volume of Griess reagent (Sigma-Aldrich, St. Louis, USA). The OD was measured at 570 nm using a microplate reader. Each test was performed in triplicate. The concentration of nitric oxide production was calculated using the following equation:

Nitric oxide production = (OD of test/OD of standard) × concentration of standard

## Statistical analysis

Data are expressed as mean ± standard error calculated over three independent experiments performed in triplicate. Statistical significance was calculated using One-way ANOVA followed by Tukey's post-hoc test. $p < 0.05$ was considered as significant. GraphPad Prism version 9 software was used for all analysis.

## RESULTS

### Determination of MIC and MBC

The antibacterial activities of the LCFS of *Lactobacillus* against the four pathogenic bacteria were determined using a broth microdilution assay. As shown in Table 1, the 10 LCFS of *Lactobacillus* showed strong antibacterial activity and inhibited *E. coli* DMST4212, *A. baumannii* DMST 2271, *S. aureus* DMST 2928, and MRSA with MIC values in the range of 25–50 mg/mL. The MBC values of these LCFS of *Lactobacillus* were >100 mg/mL. The LCFS of *Lactobacillus* T0902, T1301, and T1304 did not inhibit *S. aureus* DMST 2928 or MRSA.

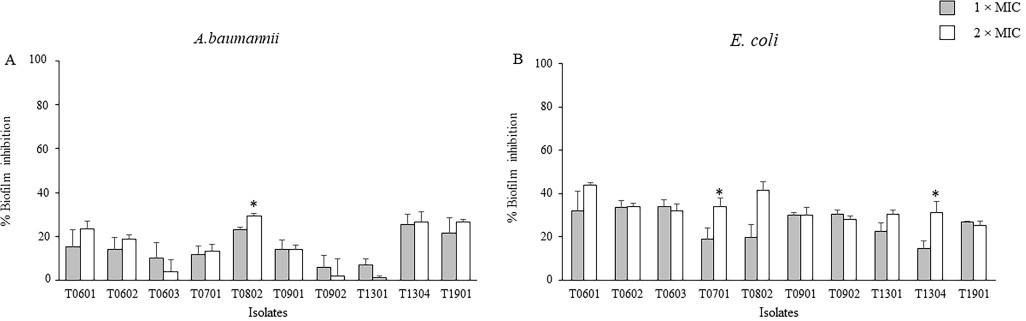

**Figure 1** **Effects of the lyophilized cell-free supernatants of *Lactobacillus* on the inhibition of biofilm formation by *A. baumannii* (A) and *E. coli* (B).** The pathogens were grown in a medium supplemented with the cell-free supernatants (CFCs) at different concentrations. CFS-free medium was used as the negative control. The relative percentage of biofilm inhibition was defined as follows: [100 − (mean A570 of treated well/mean A570 of control well) × 100]. The percent inhibition of each datum was compared with its negative control. The data are presented as mean ± standard deviation (* significant difference; $P < 0.05$).

## Reduction of biofilm formation in *A. baumannii* and *E. coli* by LCFS of *Lactobacillus*

The inhibitory activities of the LCFS of *Lactobacillus* against biofilm formation by *A. baumannii* and *E. coli* were determined using the crystal violet assay. As shown in Fig. 1 and Table S1, the concentration of the CFS tested significantly inhibited biofilm formation by *E. coli* when compared with the control. At 2× MIC, the CFS produced by the isolates T0601 and T0802 exhibited the highest inhibition (mean ± standard deviation) of 43.86% ± 1.15% and 41.35% ± 4.19%, respectively, against *E. coli* biofilm (Table S1). It has been highlighted that at 2× MIC of the supernatant of the probiotics T0701 and T1304 significantly inhibited *E. coli* biofilm formation, compared with the concentration at 1× MIC. The isolate T0802 also exhibited the highest inhibition of 29.33% ± 1.15% against *A. baumannii* biofilm. A significant difference in inhibition was observed when the bacteria were treated with 2× MIC of CFS produced by the isolate T0802 when compared with 1× MIC of the CFS. It has noticed that antibiofilm activity of the supernatant of the probiotics against both *S. aureus* and MRSA was performed. However, the results demonstrated that the supernatant did not inhibit the biofilms of both the strains.

## Activity of LCFS on the eradication of the established biofilms of *A. buamannii* and *E. coli*

The activity of the LCFS of *Lactobacillus* on the established biofilms of *A. baumannii* and *E. coli* was assessed using the crystal violet assay. As shown in Fig. 2 and Table S2, a significant decrease in the viability of mature two-day-old biofilm-grown cells of both *A. baumannii* and *E. coli* was observed after treatment with the LCFS of *Lactobacillus* at 2× MIC and 1× MIC when compared with the negative control ($P < 0.05$). The CFS from the isolate T1901 resulted in the highest eradication of 62.98% ± 3.54% and 84.34% ± 0.98% of the established biofilm of *A. baumannii* and *E. coli*, respectively. A significant difference
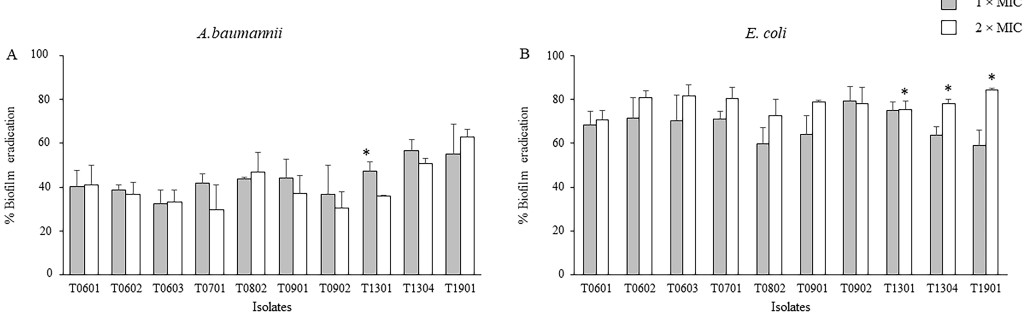

**Figure 2** **Effects of the lyophilized cell-free supernatants (LCFS) of *Lactobacillus* on the inhibition of the established biofilms of *A. baumannii* (A) and *E. coli* (B).** The bacteria were grown in a medium supplemented with glucose to produce established biofilms. The established biofilms were treated with LCFS of *Lactobacillus* at different concentrations. Cell-free supernatant-free medium was used as the negative control. The relative percentage of biofilm eradication was defined as follows: [100 − (mean A570 of treated well/mean A570 of control well) × 100]. The percent inhibition of each datum was compared with its negative control. The data are presented as mean ± standard deviation (* significant difference; $P < 0.05$).

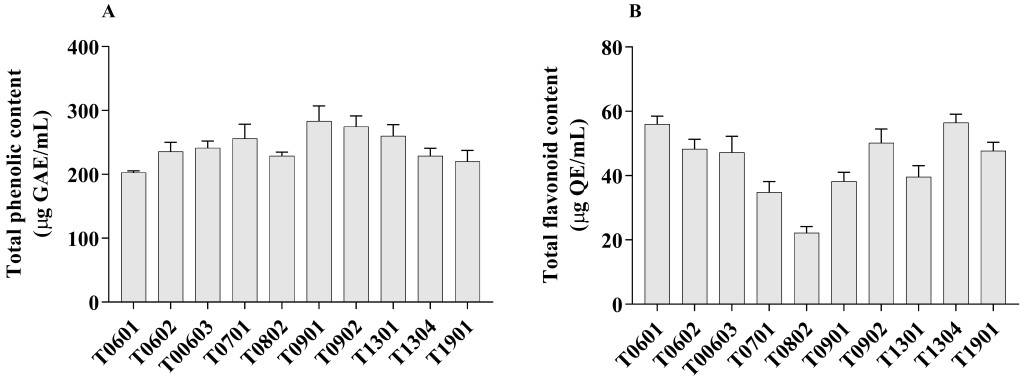

**Figure 3** **Total phenolic content and total flavonoid content of lyophilized cell-free supernatant of *Lactobacillus*.**

in the eradication was observed when the bacterial cells were treated with 2× MIC of CFS produced by the isolate T1901 when compared with 1× MIC of the CFS.

## Antioxidant activity of LCFS from *Lactobacillus*

The antioxidant activities of all isolates were evaluated by measuring the TPC, TFC, DPPH radical scavenging activity, and ABTS$^+$ radical scavenging activity (Figs. 3 and 4).

The TPC value of the LCFS of *Lactobacillus* ranged from 202.7 ± 1.42 µg GAE/g to 283.4 ± 11.91 µg GAE/g (Fig. 3A). LCFS of *L. paracasei* T0901 showed the highest TPC value (283.4 ± 11.91 µg GAE/g), followed by LCFS of *L. paracasei* T0902 (274.7 ± 8.34 µg GAE/g) and LCFS of *L. paracasei* T1302 (260.3 ± 8.69 µg GAE/g).

Values of TFC were determined in mg QE/g of lyophilized CFS of *Lactobacillus*. The TFC value of the LCFS ranged from 22.26 ± 0.94 µg QE/g to 56.60 ± 1.34 µg QE/g (Fig. 3B).
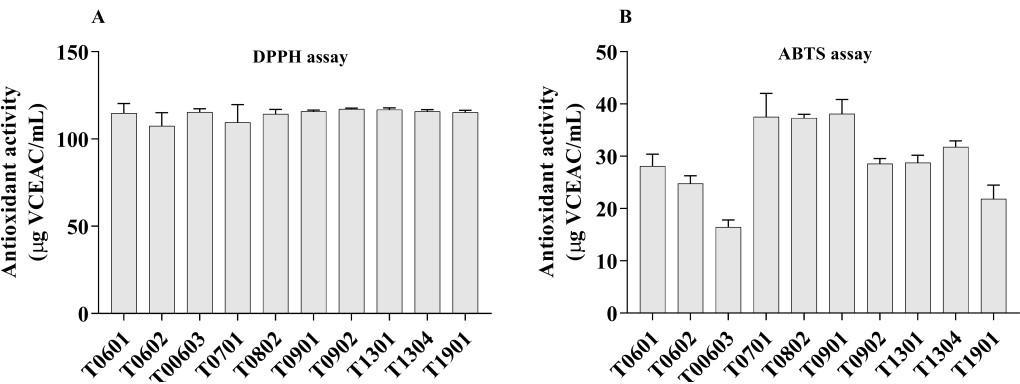

**Figure 4** Scavenging activity of lyophilized cell-free supernatant (LCFS) of *Lactobacillus* isolates, as determined by DPPH assay (A); ABTS radical scavenging activity of LCFS of *Lactobacillus* isolates (B). Values are mean ± standard error of the mean of three replicates.

LCFS of *L. paracasei* T1304 showed the highest TFC value (56.60 ± 1.34 μg QE/g), followed by LCFS of *L. paracasei* T0601 (56.03 ± 1.23 μg QE/g) and LCFS of *L. paracasei* T0902 (50.19 ± 2.15 μg QE/g).

The DPPH radical and ABTS$^+$ radical scavenging activities were used as a tool to investigate the antioxidant properties of the 10 LCFS *Lactobacillus* isolates (Fig. 4A). The results showed that all the isolates had antioxidant property.

The LCFSs of *L. paracasei* T0902 exhibited strong DPPH radical scavenging activities (117.2 ± 0.26 μg VCEAC/mL), followed by LCFS of *L. paracasei* T1301 (116.8 ± 0.53 μg VCEAC/mL) and LCFS of *L. paracasei* T1304 (115.9 ± 0.47 μg VCEAC/mL). This difference was not statistically significant ($p > 0.05$). The antioxidant activity (ABTS) of all LCFS of *L. paracasei* isolates ranged from 16.46 ± 0.67 μg VCEAC/mL to 38.1 ±1.37 μg VCEAC/mL. All of these LCFS were significantly different from each other. The LCFS of *L. paracasei* T0902 displayed the highest ABTS$^+$ radical scavenging activity (38.1 ± 1.37 μg VCEAC/mL), followed by LCFS of *L. fermentum* T0701 (37.51 ± 2.25 μg VCEAC/mL) and LCFS of *L. brevis* T0802 (37.32 ± 0.34 μg VCEAC/mL), which were not significantly different from LCFS of *L. paracasei* T0902.

## Cell viability by MTT assay

We evaluated the cytotoxicity of the 10 LCFS of *Lactobacillus* isolates in RAW 264.7 cells using MTT assays. None of these isolates produced any significant cytotoxicity in the concentration range of 5.00–118.80 mg/mL (Fig. S1). Thus, the LCFS was considered to be safe and was evaluated further.

## NO production

NO is a multifunctional mediator and plays a pivotal role in the immune response to inflammation. Results of the NO assay (Fig. 5) established that the LCFS of *Lactobacillus* showed a wide range of NO production levels. All of these isolates reduced the NO production to <10 μM (4.17 ± 1.61–8.66 ±0.23 μM) in LPS-stimulated RAW 264.7 cells when compared with untreated LPS-stimulated RAW 264.7 cells (39.89 ± 0.91 μM).

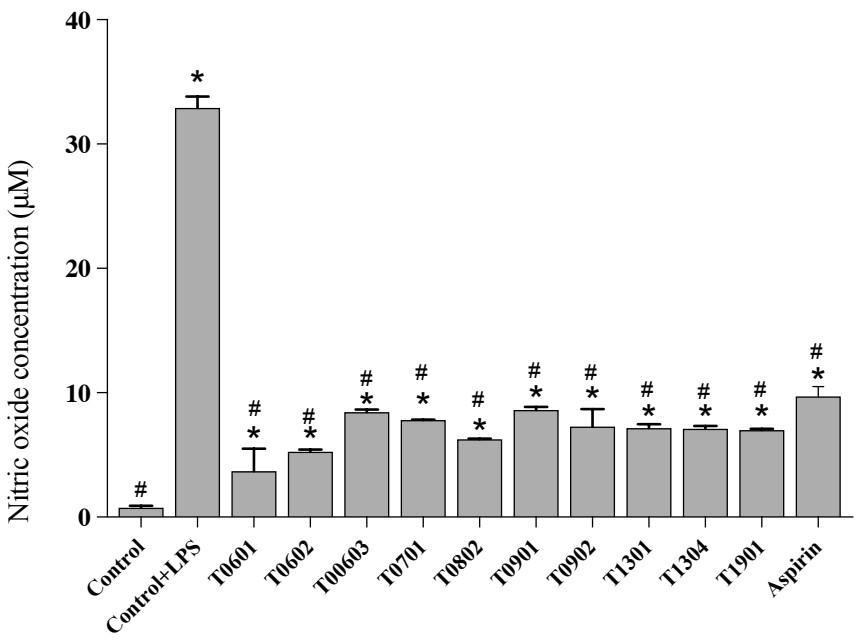

**Figure 5** **Inhibition of nitric oxide production in the lipopolysaccharide-stimulated RAW264.7 cells treated with the 10 lyophilized cell-free supernatants of *Lactobacillus* isolates and aspirin as control.** The results are presented as the mean ± standard deviation of three independent experiments ($n = 3$). * and # letters shown in the column indicate significant differences ($p < 0.05$) when compare with control, and control stimulated-LPS, respectivly.

Among the isolates, LCFS of *L. paracasei* T0601 exhibited the lowest NO production (4.17 ± 1.61 µM) in LPS-stimulated RAW 264.7 cells, followed by LCFS of *L. paracasei* T0602 (5.17 ± 0.05 µM) and LCFS of *L. brevis* T0802 (6.24 ± 0.04 µM). The NO production of aspirin-treated LPS-stimulated RAW 264.7 cells was 10.06 ± 0.50 µM and was not significantly different from that of the LCFS of *Lactobacillus* treated LPS-stimulated RAW 264.7 cells.

## DISCUSSION

Probiotics are living microorganisms that confer health benefits to the host when administered in adequate amounts. Moreover, dead bacteria, inactivated bacteria, and bacterial components can also display probiotic properties (*Plaza-Diaz et al., 2019*). Probiotics are safe, survive in the gastrointestinal tract, produce active molecules that inhibit pathogens, stimulate the immune system, and aid in the improvement of intestinal barrier function and microflora (*De Marco et al., 2018*; *Plaza-Diaz et al., 2019*). In our previous report (*Sornsenee et al., 2021*), 10 lactobacilli isolated from fermented palm sap serve as promising candidates for probiotics since they exhibit potential probiotic properties. Probiotic microorganisms, especially *Lactobacillus* species, are used as dietary supplements and capsules and in probiotic foods, beverages, and probiotic juices (*Saxelin et al., 2005*). Commercial *Lactobacillu* s strains include *L. acidophilus* NCFM, *L. acidophilus* La-5, *L. casei* Shirota, *L. casei* DN-114 001, *L. reuteri* DSM 17938, *L. rhamnosus* GG,

*L. rhamnosus* HN001, *L. rhamnosus* GR-1, *L. paracasei* F19, and *L. plantarum* 299v (*Delley et al., 2015*; *Tremblay et al., 2021*). Some *Lactobacillus* spp. are GRAS by the European Food Safety Authority (EFSA) and FDA (*Ogier & Serror, 2008*; *Plaza-Diaz et al., 2019*). The effects of these probiotics on host health have been reported in many studies (*Barzegari et al., 2020*; *Chatatikun et al., 2020*; *Hotel & Cordoba, 2001*; *Saxelin et al., 2005*; *Stefania et al., 2017*). Dead bacteria, metabolic by-products, and bacterial molecular components have also been shown to exhibit probiotic effects in various studies (*De Marco et al., 2018*; *Yang et al., 2021*). Currently, the term "postbiotic" refers to soluble components with biological activity that could be a safer alternative to the use of whole bacteria (*Tsilingiri et al., 2012*).

Antimicrobial susceptibility tests showed that all LCFS of *Lactobacillus* isolates had strong inhibitory effects on the four tested pathogens: *E. coli* DMST4212, *A. baumannii* DMST 2271, *S. aureus* DMST 2928, and MRSA. According to the results of MIC and MBC assays, the MBC/MIC ratio was more than four times that considered to be valuable as a bacteriostatic agent (*Levison, 2004*). Thus, these LCFS of *Lactobacillus* isolates are potential antibacterial agents. Our results agree with those of previous studies; for example, *Melo et al. (2016)* reported that *Lactobacillus* supernatants inhibited *S. aureus*. Other reports have shown that the lyophilized cell-free extract of *L. casei* can inhibit *E. coli*, *Salmonella typhi*, *Pseudomonas aeruginosa*, *S. aureus*, and MRSA (*Saadatzadeh et al., 2013*). Lactobacilli can produce various secondary metabolites that exhibit antimicrobial activity, such as organic acids, ethyl alcohol, short-chain fatty acids, bacteriocins, hydrogen peroxide, surfactants, and bacteriocins (*Melo et al., 2016*; *Plaza-Diaz et al., 2019*).

Biofilm-related infections are a serious clinical problem and include chronic infections. Since biofilms are not fully available to the human immune system or antibiotics, they are difficult to eradicate and control, which leads to the emergence of antibiotic-resistant strains (*Barzegari et al., 2020*; *Khairy et al., 2020*). The present study revealed that all LCFS of *Lactobacillus* isolates were able to not only inhibit pathogen biofilm formation but also eradicate mature biofilms of *E. coli* DMST4212 and *A. baumannii* DMST 2271. Probiotics can interrupt the activity of pathogens and their adhesion to surfaces. Probiotics prevent quorum sensing and biofilm formation, interfere with biofilm integrity, and eradicate biofilms by secreting antagonistic substances (*Plaza-Diaz et al., 2019*). These data, according to *Kim, Kim & Kang (2019)*, showed that *L. brevis* DF01 bacteriocin can inhibit the formation of biofilms by *E. coli* and *S. typhimurium*. Other study from *Rossoni et al. (2018)* reported that *L. fermentum* 20.4, *L. paracasei* 11.6, *L. paracasei* 20.3, and *L. paracasei* 25.4 produce bioactive substances that caused a significant reduction in *S. mutans* biofilms. Furthermore, the result similar to report from *Carvalho et al. (2021)*, *L. plantarum* showed promising results against pathogenic biofilms. Some of the bacteriocins eradicate biofilms by inducing the formation of pores on the bacterial cell surface, which leads to ATP efflux, while others exert their biological activity through proteolytic enzymes (*Okuda et al., 2013*). We consider all LCFS of *Lactobacillus* isolates to be potentially applicable for reducing the formation of biofilms and for eradicating the established biofilms of *E. coli* and *A. baumannii*.

The isolates have desirable properties as potential probiotics. During fermentation, lactobacilli can produce phenolic and flavonoid compounds as end products. The increase

in the production of these compounds during the enzymatic hydrolysis of lactobacilli during fermentation leads to an increase in their antioxidant activities (*Filannino et al., 2015*). In this study, we investigated the total phenolic and flavonoid contents of the LCFS of *Lactobacillus* isolates. All isolates contained high levels of these compounds. These findings agree with those of *Talib et al. (2019)* who reported that *Lactobacillus* spp. showed high antioxidant activities for TPC and TFC. Another study found that *L. plantarum* can produce high levels of phenolic compounds during fermentation (*Xiao et al., 2015*). The LCFS of *Lactobacillus* isolates exhibited strong DPPH and ABT·+ radical scavenging activities. Several probiotics can enhance the activity of antioxidant enzymes or modulate circulatory oxidative stress (*Mishra et al., 2015*). The CFS of *L. acidophilus, L. casei, Lactococcus lactis, L. reuteri, and Saccharomyces boulardii* could reduce oxidative damage and free-radical-scavenging rate (*De Marco et al., 2018*). *Liu & Pan, 2010*) documented that 12 *Lactobacillus* strains showed varying capabilities of DPPH radical scavenging. Thus, these results suggest that phenolics and flavonoids are the major compounds responsible for the antioxidant activities.

Inflammation is the mark of many inflammatory disorders such as chronic peptic ulcer, Crohn's disease, and infections. The intestinal immune system has developed distinct mechanisms to dampen mucosal immunity and to optimize the response against microbiota. NO is a multifunctional mediator and plays an essential role in the immune response to inflammatory activity. Normal NO production in the phagocytes is beneficial for host defense against pathogens and cancer cells (*Abdulkhaleq et al., 2018*). Proinflammatory cytokines are commonly induced by the LPS cell-wall component of gram-negative bacteria. In this study, the LCFS of *Lactobacillus* isolates showed low levels of NO production. The supernatant did not exhibit any cytotoxic activity against the RAW 264.7 cells. Recently, there have been a few studies on the anti-inflammatory activity of the CFS of probiotics. *Kang et al. (2021)* observed that *Bifidobacterium bifidum* MG731, *B. lactis* MG741, and *L. salivarius* MG242 showed low NO production. In another report, the CFS of *L. acidophilus* and *L. rhamnosus GG* showed anti-inflammatory properties and modulated the inflammatory response (*Maghsood et al., 2018*). Thus, reduced NO production by the LCFS of *Lactobacillus* isolates may be due to the downregulation of inducible NO synthase, the main mediator of various chronic inflammatory diseases (*Oh et al., 2012*).

Exploiting the LCFS of *Lactobacillus isolates* in the preparation of probiotic products is an innovative approach and has the potential to replace the living probiotic cells.

## CONCLUSIONS

The present study revealed that the 10 LCFS of *Lactobacillus* isolates exhibited antibacterial activity, reduced the formation of biofilms, and eradicated the established biofilm. These supernatants contain phenolic and flavonoid compounds and display antioxidant and anti-inflammatory activities in RAW 264.7 cells. Therefore, they are promising novel postbiotic candidates for use in functional foods and pharmaceuticals. Further research to elucidate the precise molecular mechanisms of action of probiotics is warranted.

## ACKNOWLEDGEMENTS

The authors thank the Research Institute for Health Sciences Walailak University, School of Allied Health Sciences, Walailak University, for providing the required laboratory instruments.

### Funding

The authors received no funding for this work.

### Competing Interests

The authors declare there are no competing interests.

### Author Contributions

- Phoomjai Sornsenee conceived and designed the experiments, performed the experiments, analyzed the data, prepared figures and/or tables, authored or reviewed drafts of the paper, and approved the final draft.
- Moragot Chatatikun, Kantapich Kongpol, Sasirat Sohbenalee, Supawita Pruksaphanrat and Amron Mudpan performed the experiments, prepared figures and/or tables, and approved the final draft.
- Watcharapong Mitsuwan and Nateelak Kooltheat analyzed the data, prepared figures and/or tables, and approved the final draft.
- Chonticha Romyasamit conceived and designed the experiments, performed the experiments, prepared figures and/or tables, authored or reviewed drafts of the paper, and approved the final draft.

### Data Availability

The raw measurements are available in the Supplementary Files.

### Supplemental Information

Supplemental information for this article can be found online at http://dx.doi.org/10.7717/peerj.12586#supplemental-information.

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
