# Peer review of "Lyophilized cell-free supernatants of Lactobacillus isolates exhibited antibiofilm, antioxidant, and reduces nitric oxide activity in lipopolysaccharide-stimulated RAW 264.7 cells"

_PeerJ, doi:10.7717/peerj.12586_

## Round 0.1 · original submission · Major Revisions

Reviewers have raised some serious concerns and shortcomings in the study. MAJOR revision is suggested, which requires substantial and thorough revision to appreciate the quality of the manuscript for publication in PeerJ. Therefore, authors are requested to revise their manuscript in light of the reviewers' comments. Please justify and discuss all questions raised by the reviewers and resubmit accordingly.

Reviewer 1 ·

Basic reporting

Clear and professional English.

Experimental design

Sound.

Validity of the findings

Valid.

Additional comments

The authors studied the antibiofilm, antioxidant, and antiinflammatory activities of lyophilized cell-free supernatants of lactobacillus isolates. The manuscript is scientifically sound. Methods and conclusions are appropriate.

·

Basic reporting

The study is a interesting approach to probiotic properties of lactobacilli against biofilms.

The english is clear with sufficiente background and references in Itroduction section.

It is missing statistical analysis in the Materials and Methods section and no limitations are recognized by the authors. They started with four pathogenic bacteria but only evaluate antibiofilm properties with the two Gram-negative bacteria.

Bold statements are done in Discussion section and repetition with Introduction section was made.

Experimental design

The manuscript is within aims and scope of the journal.

The research question is well defined and relevant.

The Materials and Methods and Discussion sections need major revisions by the authors.

Validity of the findings

The findings are worth to be publish in the journal. However, major revisions are needed before acceptance.

The conclusions showed bold statements and no limitations were recognized by the authors.

Additional comments

I did all my comments and suggestions in the annotated PDF file of the manuscript in attachments.

Reviewer 3 ·

Basic reporting

Authors should improve the literature background.

Experimental design

Nitric oxide reduction is not enough to prove anti-inflammatory effects. Authors should provide more evidence regarding anti-inflammatory activities by ELISA, for instance.

Validity of the findings

Statistics should be presented. How was the data processed? Which statistical test was used? It should be provided.

Additional comments

“Lyophilized Cell-Free Supernatants of Lactobacillus Isolates Exhibited Antibiofilm, Antioxidant, and Anti- inflammatory Activities in Lipopolysaccharide stimulated RAW 264.7 cells” is an interesting paper that brings new evidence of the biotechnological potential use of postbiotics in vitro. This topic is current and attractive. However, it needs some improvements before being considered for publication.

Title
- Why did the authors affirm that the postbiotics had anti-inflammatory activity? Nitric Oxide alone cannot prove that. This way, the title should be changed.
Abstract
- Some parts of the results contain methodology steps. It should be removed. Also, the conclusion should be more specific and focused.
Introduction
- Authors should present this reference in the introduction:
- Xu X, Qiao Y, Peng Q, Shi B, Dia VP. Antioxidant and Immunomodulatory Properties of Partially purified Exopolysaccharide from Lactobacillus Casei Isolated from Chinese Northeast Sauerkraut. Immunol Invest. 2021 Jan 8:1-18. doi: 10.1080/08820139.2020.1869777. Epub ahead of print. PMID: 33416001.
Materials and Methods
- Authors should evaluate the production of key cytokines (such as TNF-α and IL-10) to prove that they had any anti-inflammatory activity. It is not possible to conclude that they have anti-inflammatory activities just performing Nitric Oxide by the Griess reaction.
- It is not clear why the authors chose 16 h to perform the Griess reaction.
- Authors should include how they evaluated the data. The statistical analysis should be added in the Materials and Methods section.
Results
- Table 1 – The name of the four bacteria used in this assay should be added in the table legend.
- Figure 1 – What are the controls for this experiment? What the authors used as a positive control of antibiofilm inhibition activity? Also, the quality of the image should be increased.
- Figure 5- There is no statistical difference between the tested groups? It is not clear in the graphic.
Discussion
- Authors should explore further why postbiocts were more effective regarding eradication than inhibition.
- Again, it is not possible to affirm that the postbiotics have anti-inflammatory effects just by evaluating the NO production. How about the key pro-inflammatory cytokines?

Conclusion
- The conclusion should be more focused and be clear about what this paper brings of novelty to the literature.

---

## Round 0.2 · accepted · Accept

The manuscript is significantly improved by the authors and now can be accepted in its current form.

·

Basic reporting

The manuscript was rectified following the Reviewer's comments and suggestions.

Experimental design

The manuscript was rectified following the Reviewer's comments and suggestions.

Validity of the findings

The manuscript was rectified following the Reviewer's comments and suggestions.

Additional comments

Congratulations to the authors.

Reviewer 3 ·

Basic reporting

The paper is clear.

Experimental design

The design is enough.

Validity of the findings

The statistics are correct.